# Growth and Development of Leaf Vegetable Crops under Conditions of the Phytotechnical Complex in Antarctica

Gayane G. Panova [1,*], Andrey V. Teplyakov [2], Anatoliy B. Novak [2], Margarita A. Levinskikh [3], Olga R. Udalova [1], Galina V. Mirskaya [1], Yuriy V. Khomyakov [1], Dmitry M. Shved [3], Evgeniy A. Ilyin [3], Tatiana E. Kuleshova [1], Elena V. Kanash [1] and Yuriy V. Chesnokov [1]

[1] Agrophysical Research Institute (AFI), 195220 Saint-Petersburg, Russia; udal59@inbox.ru (O.R.U.); galinanm@gmail.com (G.V.M.); himlabafi@yandex.ru (Y.V.K.); www.piter.ru@bk.ru (T.E.K.); ykanash@yandex.ru (E.V.K.); yuv_chesnokov@agrophys.ru (Y.V.C.)

[2] Arctic and Antarctic Research Institute (AARI), 199397 Saint-Petersburg, Russia; andrey-valerjevich@ya.ru (A.V.T.); ntolich@mail.ru (A.B.N.)

[3] Institute of Biomedical Problems (IBMP), 123007 Moscow, Russia; ritalev@imbp.ru (M.A.L.); d.shved84@gmail.com (D.M.S.); pag_as@mail.ru (E.A.I.)

* Correspondence: gaiane@inbox.ru; Tel.: +7-(812)-535-79-09

**Abstract:** Ensuring the technical and technological possibility of regularly obtaining fresh, high-quality plant production in Antarctic stations is an urgent task of our time. This work is devoted to studying the growth and development of leaf vegetable crops and the main quality indicators of their edible parts when grown in the phytotechnical complex greenhouses at the "Vostok" Antarctic station and at the agrobiopolygon of the Agrophysical Research Institute (AFI). The plants, belonging to 13 varieties of 9 types of leaf vegetable crops (arugula, garden cress, cabbage, mustard, leaf radish, leaf lettuce, amaranth, dill, parsley leaf)**,** were studied during five growing seasons at the "Vostok" station and at the AFI agrobiopolygon under controlled conditions (control). The experimental data obtained demonstrate the high productivity of the phytotechnical complex for most of the investigated crops per unit of useful area, with lower costs of electricity and water consumption per unit of production compared with a number of greenhouses at foreign Antarctic stations and greenhouse complexes with controlled conditions located on other continents. Lettuce crops were the most adapted to the growing conditions at the Antarctic station "Vostok". They did not differ in their evaluated characteristics from the control. All other investigated crops, while not differing in their development rate and quality, had statistically significant (16–61%) decreases in their yield per 1 m$^2$ per year. This may demonstrate the difference in the "genotype–environment" interaction in plants grown at the Antarctic station and AFI agrobiopolygon, probably due to the different barometric pressure and partial pressure of oxygen at the two locations. The positive psychological effects of the greenhouses were identified along with nutritional and other qualities of the plants.

**Keywords:** Antarctic station "Vostok"; AFI agrobiopolygon; plant growing light equipment; thin-layer panoponics; leafy vegetable; "genotype–environment" interaction; plant production; yield; quality

## 1. Introduction

Human needs with respect to the natural and agricultural ecosystem are especially important in regions with extreme climate, such as Antarctica. The natural growth of plants, common for other continents, is impossible in most of the territories of Antarctica [1]. Along with providing a number of vitamins, proteins, macro- and microelements, and other human nutrition ingredients, vegetable plants boost human immunity and improve health in general [2]. They also have positive psychoemotional effects, which are beneficial under the extreme conditions of the Antarctic [3].

In this regard, at most Antarctic stations belonging to different countries, from 1946 to the present, greenhouses were established for the cultivation of various vegetable, medici-

nal, and ornamental crops. Most of these greenhouses were later closed and dismantled for various reasons [1]. A few publications have reported solutions for the arrangements of greenhouses in the Antarctica, ranging from plants-growing light equipment in the station premises to greenhouses housed in specialized plant-growing construction or building rooms with controlled microclimate conditions [1,4]. Most of the current plant growing systems are located at coastal stations in the northwestern part of Antarctica (Antarctic Peninsula and adjacent islands), which have a relatively mild climate. A number of greenhouses are also located at stations situated along the perimeter of the eastern and southern parts of the mainland. Only two stations are operational all year round, the U.S. and Russian stations, which are located inside the continent at the South Pole and in the region of the Earth's cold pole, respectively. These locations have the most extreme conditions for living organisms.

There are different types of plant-growing systems at the polar stations. The U.S. Amundsen–Scott South Pole station houses one of the currently operating greenhouses—the South Pole Food Growth Chamber (SPFGC) (created in 2004)—which is located in several rooms in the station building. One room is for growing plants, while other associated service rooms have blocks for preparation and storage of nutrient solutions and for plant growth monitoring and microclimate-regulating systems. There is also an observational relaxation room for station employees, which is separated from the plant production room with a glass wall [4,5]. Plants are grown in two-tier and single-tier plant growing equipment under artificial light using the nutrient film (NFT) and deep flow (DFT)hydroponic techniques. The process of growing plants is quite energy-consuming. For example, plant production over an area of 22.77 $m^2$ has an electricity consumption of 12.8 $kWh^{-1}$ $m^{-2}$, including the operation of light sources, which require 9.5 $kWh^{-1}$ $m^{-2}$. The amount of plant biomass produced per day and electricity consumed is 0.01 kg $kWh^{-1}$ $d^{-1}$ [5].

The most modern EDEN ISS greenhouse (created in 2018), which serves as a prototype for future space stations on the Moon or Mars, is located near the German Neumayer III station in Antarctica and is a stand-alone, container-type facility [6]. It consists of two 20 ft-high customized cube shipping containers connected to each other and placed on top of a raised platform. Inside, the facility is divided into three sections: an airlock/cold porch small room providing storage and a small air buffer to limit the entry of cold air into the facility; a service section; and an exploration greenhouse or green section [3,6,7].

Inside the green section, there are multi-level growth racks in which the plants are grown under controlled conditions. LED light sources are used, with the spectrum consisting mainly of blue (~450 nm) and red (~650 nm) wave bands and small portions of other wavelengths. The light intensity when growing leafy vegetables is 330 μmol $m^{-2}$ $s^{-1}$ at canopy level. Plants are cultivated using the methods of aeroponics or aerohydroponics, and the useful area for plant growth is 12.5 $m^2$ [6,7].

The interior of the EDEN International Space Station greenhouse is similar to that of the King Sejong Institute in the Republic of Korea, created in 2010, where the plants are also placed in separate containers [4]. Inside their green section, there are multi-level plant-growth racks along the walls where leaf vegetable crops, tomatoes, cucumbers, watermelon, and other plants are grown under artificial light from fluorescent or LED lamps in a controlled environment. Plants are cultivated using the method of ebb and flow [4].

There is also a greenhouse, established in 2014/2015, at the Chinese station "The Great Wall of China", where plants are grown in a specialized greenhouse made of steel structures and translucent plexiglass (polymethyl methacrylate) material [1]. Such a greenhouse has a gateway and a room for microclimate regulation. Plants are grown on tiered structures using the hydroponic method of ebb and flow [1,4]. Illumination is carried out using natural sunlight, high-pressure sodium (HPS) lamps, and LED lamps [4].

Tiered plant growing systems, located in separate station rooms, are used at the South Korean station Jang Bogo, the Japanese station Syowa, and the New Zealand station Scott

Base [4]. Unfortunately, detailed information on the productivity of the greenhouses and the costs of plant production at most stations is not yet available in the open access literature.

At the mentioned U.S., German, Chinese, South Korean, and other Antarctic stations, the greenhouses are located in separate rooms or structures, mostly 100–400 m away from the staff accommodations. As noted in several reports [1,4–7], overcoming this distance in frosty air represents a marked psychological and physical discomfort for people, as well as limited access to greenhouses in separate rooms. The main advantages and disadvantages of the technical, technological, economic, and psychoemotional aspects of this arrangement of greenhouses are listed in the work [2].

Russian scientists from the Agrophysical Research Institute (AFI), Arctic and Antarctic Research Institute (AARI), and Institute of Biomedical Problems (IBMP), together with Russian polar researchers, considered the possibility of creating and placing greenhouses directly within the researcher accommodations.

The possibility of growing plants in a closed hermetic volume with humans [8,9] was proven through a series of experiments carried out at IBMP since 1968. Since 1932, AFI has developed original technologies for intensive year-round cultivation of agricultural, medicinal, and ornamental plants under the controlled conditions of intense light culture [10]. These were based on knowledge of the patterns of interaction between plants and the environment in a regulated agroecosystem [10–13], making it possible to predict the successful implementation of the above-mentioned plant growing systems at Antarctic stations.

In 2019, based on the AFI agrobiopolygon, the phytotechnical complex greenhouse for year-round cultivation of plants up to 35–40 cm high was developed, built, and tested. Thin-layer panoponics [10–12] and original LED lamps, giving light in a spectrum close to sunlight in the PAR region, were implemented in the phytotechnical complex greenhouse. This complex was delivered for testing to the "Vostok" Antarctic station by the 65th Russian Antarctic Expedition of AARI.

It should be noted that the "Vostok" Antarctic station, due to its geographical location and extreme life conditions, is considered to be an analogue of a long-term-inhabited space base on the Moon, and the information obtained makes it possible to predict the behavior of people and their physical and emotional states under space base conditions.

The hypothesis of this research is that creating similar light and air conditions and using the same nutrient-providing techniques in the phytotechnical complex greenhouses at the "Vostok" Antarctic station and the control AFI agrobiopolygon will reveal the diversity of reaction and adaptability level of test crops under the unique conditions of the two locations, which differ in barometric pressure and the partial pressure of oxygen. It is also expected that the psychological and emotional reactions of polar researchers to plants during their stay at the Antarctic station will be positive and similar to that of cosmonauts and other testers who have stayed isolated for a long time under extreme environmental conditions.

The purpose of this work was to study the growth and development of leaf vegetable crops and the main quality indicators of their edible parts when grown in the phytotechnical complex greenhouses at the "Vostok" Antarctic station and at the agrobiopolygon of the Agrophysical Research Institute.

## 2. Materials and Methods

The studies were carried out at the Russian Antarctic station "Vostok" (78°27′ S 106°52′ E) and at the agrobiopolygon of AFI under controlled microclimate conditions (Saint Petersburg, Russia (60°0′2.27″ N (60.000627) 30°23′1.32″ E (30.3837)).

The object of the research was leafy vegetable crops: arugula *Eruca sativa* Mill. cv. Gourmet, cv. Barokko, garden cress *Lepidium sativum* L. cv. Aghur, cv. Vesenny, Chinese cabbage *Brassica rapa* subsp. *pekinensis* (Lour.) Hanelt cv. Vesnyanka, Japanese cabbage *Brassica rapa* ssp. *Nipposinica* (L.H.Bailey) Hanelt cv. Mizuna Green, mustard *Brassica juncea* (L.) Czern. cv. Red Giant, leaf radish *Raphanus sativus* var. *Oleiformis* Pers. cv. Rax, amaranth *Amaranthus* L. cv. Bagryanets, leaf lettuce *Lactuca sativa* L. cv. Typhoon, cv. Lollo Rossa,

dill *Anethum graveolens* L. cv. Russian giant, and parsley leaf *Petroselinum crispum* var. *crispum* Mill. cv. Bogatyr. Plant seeds were received from the seed joint stock company "Sortsemovoshch" (Saint Petersburg, Russia).

Plants were grown in phytotechnical complex greenhouses located at the Russian Antarctic station "Vostok" in one of its living rooms with an area of 12 m$^2$ and at the AFI agrobiopolygon in a specialized production test room with controlled microclimate conditions.

The phytotechnical complex greenhouse is a piece of automated, two-tier, plant-growing light equipment with a useful area of 1 m$^2$. The plants are grown on a thin-layer analogue of soil (thin-layer panoponics), developed at AFI and patented [14]. A reusable hydrophilic material made of polyethylene terephthalate is placed into the trays of the phytotechnical complex. Seeds of plants are seeded into a 1-mm layer of suspension, based on Cambrian clay, on the surface of the material. Through the flat slotted capillaries, the latter provides plant root systems with a nutrient solution circulating over the tray bottom [10–12]. As a substitute for sunlight in the phytotechnical complex greenhouse, LED lamps, SL-P-80f of GREENTECH LLC (St. Petersburg, Russia), are used with a spectrum optimized for growing crops and with the possibility of varying the intensity of the light flux during the growing season.

Light sources modulating the sunlight spectrum in the photosynthetically active radiation (PAR) were used. The white LEDs included in the lamps with modified secondary optics using a polymer phosphor (know-how) with a color temperature of 4000 K simulated morning sunlight, when the sun is at an angle of about 20 degrees relative to the horizon.

The emission spectrum of LED light sources used in the phytotechnical complex greenhouse, measured with the PG200N Uprtek Sunlike PAR spectrometer, is shown in Figure Figure 1. The light period was 14 h. The air temperature in the plant growth zone was kept at +22 °C–+24 °C during the light period and +18° C–+20 °C during the dark period. Air humidity was 60–70%. Plant life-support systems worked automatically.

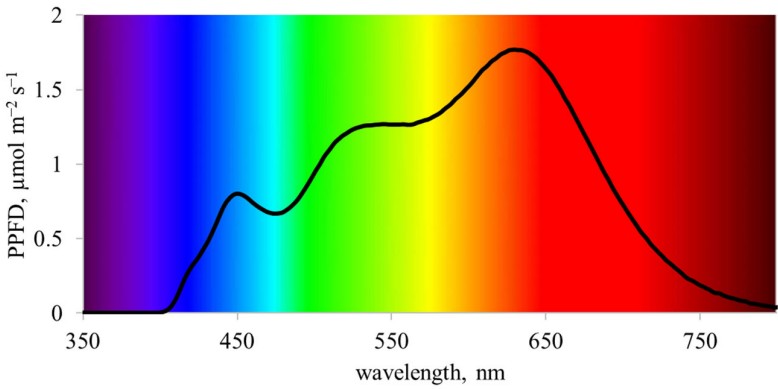

**Figure 1.** Sun-like emission spectrum of LED light source used in phytotechnical complex greenhouse.

Sowing seeds of each crop was carried at the density of 1 g/0.02 m$^2$ (repetition), which was repeated in each growing season twice for most cultivated crops, for arugula-three times. Each crop was grown over five growing seasons during 2020–2021.

Modified nutrient Knop's solution was used (Table 1), having pH 6.0 and electrical conductivity (EC) of 1.0 mS cm$^{-1}$. The frequency of supplying fresh solution to trays with plants was 6 times per day, with regular intervals.

**Table 1.** The content of macro- and microelements in the nutrient solution for leafy vegetable crops grown in a phytotechnical complex greenhouse at the "Vostok" Antarctic station.

| Content of the Element in the Nutrient Solution | | | | | | | | | | | |
|---|---|---|---|---|---|---|---|---|---|---|---|
| $NH_4$, mmol | K, mole | Ca, mole | Mg, mole | $NO_3$, mole | $SO_4$, mole | P, mole | Fe, mmol | Mn, mmol | Zn, mmol | B, mmol | Cu, mmol |
| 1.78 | 0.350 | 0.340 | 0.100 | 0.885 | 0.103 | 0.140 | 1.780 | 0.789 | 0.070 | 0.468 | 0.080 |

At the end of each growing season, the raw mass and dry mass of the edible part of the plants were determined. The values of quality and safety indicators of plant production were evaluated using standard and generally accepted methods [15–20]. The dry matter content was determined using the thermostatic-weight method [15]. Analysis for ascorbic acid (vitamin C) was carried out with high-performance liquid chromatography [16]. The nitrate content was determined using the ionometric method [17]. For raw ash and macro- and microelement analysis (zinc, copper, iron, etc.), atomic absorption spectrometry (AAS) after microwave digestion [18–20] was employed.

A psychological survey, prepared and conducted by authors from the IBMP, was administered to polar researchers at the beginning and end of their wintering period. The survey was carried out in order to determine the psychological value of the greenhouse for Vostok station personnel, evaluating their expectations and attitudes towards the presence of plants at the station.

The subjects signed informed consent forms for participation in all medical and psychological studies at Vostok station. The studies were approved by Bioethics Committee of the Institute of Biomedical Problems, Russian Academy of Sciences.

*Statistical Analysis*

Statistical processing of the results was performed using Excel 2013 and Statistica12 (Stat-Soft, Inc., Tulsa, OK, USA). The mean values of the studied parameters and their confidence intervals (CI) M ± CI were determined. The data were analyzed using one-wayanalysis of variance (ANOVA) followed by Duncan's multiple-range test to determine the significance of differences between mean values. Differences between options were considered significant at $p \leq 0.05$. When assessing the content of micronutrients in leafy vegetable crops grown during 5 growing seasons at the Russian Antarctic "Vostok" Station, yield variability was assessed using the coefficient of variation CV [21].

## 3. Results

### 3.1. Growth, Development, and Productivity of the Plants' Edible Parts

Due to the low air humidity (18–25%) and reduced partial pressure of oxygen (95–100 mm Hg) in the rooms at the "Vostok" Antarctic station, the growing conditions inside the phytotechnical complex were optimized, providing favorable temperature and air humidity values for cultivated leafy vegetable crops. The total raw mass yield of plant edible parts per year from the usable growing area of 1 m$^2$ averaged 36.5 kg. The total water consumption of the plants averaged 36.2 L/kg of edible part raw mass per year from the usable area of 1 m$^2$ in the phytotechnical complex. This value includes the average consumption of nutrient solution of 19.8 L/kg of edible part raw mass. The amount of electricity used per 1 m$^2$ per year was 113.2 kW/kg of edible part raw mass, including that used for illumination—89.6 kW/kg.

In parallel with the experiments at the "Vostok" station, the same crops were grown in a complete analogue of the phytotechnical complex at the agrobiopolygon with controlled microclimate conditions in the Agrophysical Research Institute (AFI), where the most optimal environment for growing plants has been established [11].

The photo shows green crops grown in the phytotechnical complex at "Vostok" station and at the AFI agrobiopolygon (Figure 2).

Based on the productivity of each crop grown per year at the phytotechnical complex, calculated while taking into account the experimental data on the average yield of edible parts per 1 m$^2$ for the growing season and the number of such periods per year, lettuce crops turned out to be the most adapted to the growing conditions at the "Vostok" Antarctic station. The characteristics of growth, development, biomass, and yield per 1 m$^2$ per year of the edible parts of these lettuce cultivars did not differ from those grown at the AFI agrobiopolygon (Table 2). Similar results were observed in [22] when comparing these indicators in lettuce grown in a container-type greenhouse at the Neumayer III Antarctic research station to that grown in climate chambers for growing plants in Europe.

**Table 2.** Yield of leafy vegetable crops grown in phytotechnical complex with the technology of thin-layer panoponics at the Russian Antarctic station "Vostok" and at the agrobiopolygon of the AFI.

| Indicators | Arugula *Eruca sativa* (Mill.) Thell., Cultivars | | Garden Cress *Lepidium sativum* L., Cultivars | | Chinese Cabbage *Brassica rapa* subsp. *pekinensis* (Lour.) Hanelt cv. Vesnyanka | Japanese Cabbage *Brassica rapa* ssp. *nipposinica* (L.H.Bailey) Hanelt cv. Mizuna Green | Mustard *Brassica juncea* (L.) Czern. cv. Red Giant | Leaf Radish *Raphanus sativus* var. *Oleiformis* Pers. cv. Rax | Amaranth *Amaranthus* L. cv. Bagryanets | Leaf Lettuce *Lactuca sativa* L., Cultivars | | Parsley Leaf *Petroselinum crispum* var. *cripum* Mill. cv. Bogatyr | *Dill Anethum graveolens* L. cv. Russian Giant |
|---|---|---|---|---|---|---|---|---|---|---|---|---|---|
| | Gurman | Barokko | Aghur | Vesenny | | | | | | Typhoon | Lollo Rossa | | |
| | Green mass yield per year, kg/m$^2$ | | | | | | | | | | | | |
| Antarctic station "Vostok" | 42.8 ± 4.0 [b] | 39.7 ± 2.3 [b] | 34.2 ± 3.4 [b] | 35.1 ± 2.4 [b] | 77.8 ± 11.0 [b] | 51.1 ± 4.6 [b] | 49.1 ± 4.3 [b] | 52.8 ± 9.3 [b] | 27.8 ± 3.9 [a] | 73.5 ± 5.3 [a] | 87.7 ± 11.3 [a] | 10.9 ± 1.9 [b] | 9.8 ± 0.4 [b] |
| Agrobiopolygon of the AFI | 54.2 ± 5.1 [a] | 51.2 ± 2.9 [a] | 46.3 ± 4.5 [a] | 47.4 ± 3.2 [a] | 112.9 ± 15.9 [a] | 109.1 ± 9.8 [a] | 73.2 ± 6.4 [a] | 69.8 ± 12.2 [a] | 33.2 ± 4.6 [a] | 73.8 ± 5.2 [a] | 86.9 ± 10.4 [a] | 26.4 ± 4.3 [a] | 25.2 ± 4.8 [a] |
| Productivity ratio, "Vostok"/AFI % | 79/100 | 78/100 | 74/100 | 74/100 | 69/100 | 47/100 | 67/100 | 76/100 | 84/100 | 100/100 | 101/100 | 41/100 | 39/100 |

Note: The table shows the average values and confidence interval of plant yield (M ± CI) at a 95% probability level. Data are presented as the means of five experiments with three biological replications per variant for arugula *Eruca sativa* (Mill.) Thell., cvs. Gurman, Barokko, and the means of five experiments with two biological replications per variant for other studied crops. Values in columns followed by different letters (a,b) are significantly different at $p \leq 0.05$, as determined by Duncan's multiple range test.

All other crops, showing the same rate of development, had significant decreases in yield per year per 1 m$^2$ (by 16–61%) compared to those at the AFI agrobiopolygon. There were no differences in growing conditions (temperature and humidity of the air, spectrum and intensity of the light flux, and nutrition) between the growing systems at the "Vostok" station and at the AFI agrobiopolygon. Therefore, observed differences in plant growth were likely associated with differences in the "genotype–environment" interaction, in particular, those caused by the differences in barometric pressure (62.4 kPa and 101.5 kPa) and partial pressure of oxygen (95–100 mm Hg and 150–160 mm Hg) at the "Vostok" station and the AFI agrobiopolygon, respectively. This assumption is based on the work of [23], which reported a pronounced tendency towards a decrease in growth morphometric and weight indicators of plants with a decrease in atmospheric pressure. The authors suggested the solution to this problem would be choosing plants that are more resistant to changes in atmospheric pressure.

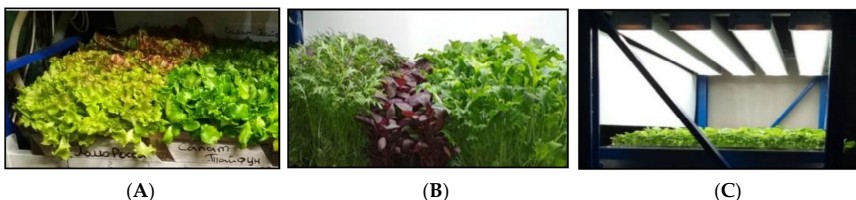

(**A**)             (**B**)             (**C**)

**Figure 2.** Leafy vegetable crops at various stages of development in the phytotechnical complex at the station "Vostok" (**A**,**B**) and at the agrobiopolygon of the Agrophysical Research Institute (**C**).

In our experiments, judging by the absolute values, Chinese cabbage *Brassica rapa* subsp. *pekinensis* (Lour.) Hanelt cv. Vesnyanka and two studied lettuce cultivars dominated in terms of the yield of raw (Table 2) and dry mass (Figure 3). The least adapted to the conditions of the "Vostok" station among the studied plants were dill *Anethum graveolens* L. cv. Russian giant and leaf parsley *Petroselinum crispum* var*. Crispum* Mill. cv. Bogatyr.

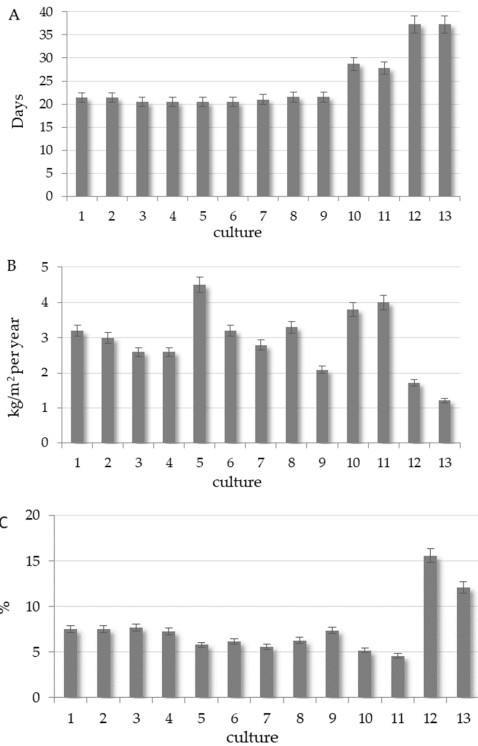

**Figure 3.** Indicators of growth and development of the studied vegetable crops grown in a phytotechnical

complex at the Russian Antarctic station "Vostok" (five growing seasons). (**A**)—average growing season before cutting; (**B**)—dry mass yield; (**C**)—content of dry matter; 1—Arugula *Eruca sativa* (Mill.) Thell., cv. Gurman; 2—Arugula *Eruca sativa* (Mill.) Thell., cv. Barokko; 3—Garden cress *Lepidium sativum* L. cv. Aghur; 4—Garden cress *Lepidium sativum* L. cv. Vesenny; 5—Chinese cabbage *Brassica rapa* subsp. *pekinensis* (Lour.) Hanelt, cv. Vesnyanka; 6—Japanese cabbage *Brassica rapa* ssp. *Nipposinica* (L.H.Bailey) Hanelt cv. Mizuna Green; 7—Mustard *Brassica juncea* (L.) Czern. cv. Red Giant; 8—Leaf radish *Raphanus sativus* var. *Oleiformis* Pers. cv. Rax; 9—Amaranth *Amaranthus* L. cv. Bagryanets; 10—Leaf lettuce *Lactuca sativa* L. cv. Typhoon; 11—Leaf lettuce *Lactuca sativa* L. cv. Lollo Rossa; 12—Parsley leaf *Petroselinum crispum* var. *crispum* Mill. cv. Bogatyr; 13—Dill *Anethum graveolens* L. cv. Bars show the average values and confidence interval (M ± CI) at a 95% probability level.

It should be noted that the estimated values of the plant development and growth indicators were fairly even in repetitions; the coefficients of variation were mainly in the range of 5.0–16.2% and did not exceed 27.0% (Table 2).

When comparing the daily productivity of lettuce and other green crops at the "Vostok" phytotechnical complex and at greenhouses at the Antarctic stations of other countries, as well as the costs of electricity and water consumption (including nutrient solutions) per unit of production (Table 3), it can be seen that in the AFI phytotechnical complex, a similar yield of edible parts per unit of time was formed at lower costs of electricity and water consumption.

**Table 3.** Productivity per unit of time and electricity and water consumption per unit of plant production in the phytotechnical complex at the Russian Antarctic station "Vostok" and greenhouses at the German Neumayer III station * and the U.S. Amundsen–Scott South Pole station **.

| Name of the Antarctic Station/Name of the Greenhouse | Greenhouse Productivity per Day, kg $(m^{-2} * d^{-1})$ | Method of Growing in the Greenhouse | Energy Consumption per Unit of Plant Production, kW kg$^{-1}$ from 1 m$^2$ | Water Consumption per Unit of Plant Production, L/kg from 1 m$^2$ | Links to Literary Sources |
|---|---|---|---|---|---|
| Antarctic station "Vostok"/phytotechnical complex 2020–2021 | 0.100 (leafy vegetable crops) | Thin layer panoponics | 113.2 | 36.2 | original data |
| German Neumayer III station/EDEN ISS greenhouse in 2018 * | 0.089 * (leafy vegetable crops. including salad. cucumber. tomato) | Aeroponic-hydroponic | 205.0 * | no data available | [6,22] |
| Amundsen-Scott South Pole station/The South Pole Food Growth Chamber (SPFGC) ** | 0.077 ** (leafy vegetable crops) | NFT and DFT hydroponic | 160.3 (281 kWh·d$^{-1}$ × 365 d/22.77 m$^2$/28.1 kg·m$^2$) ** | 95.5 ** | [3] |

Note: *—data are taken from publications or obtained by calculating the latter [6,22]; **—from [3].

Thus, a comparison of the obtained and published data presented in Table 4 shows that the productivity of the phytotechnical complex per unit time at the "Vostok" Antarctic station exhibited a trend towards higher values than that at the German Neumayer III Antarctic station and the U.S. Amundsen–Scott South Pole Antarctic station, i.e., by 11% and 23%, respectively. Additionally, the consumption of electrical energy per unit of obtained plant production at "Vostok" was 1.8 times lower than at the greenhouse at the German station and 1.4 times lower than at the U.S. station, and water consumption was 2.6 times lower than at the U.S. stations. The choice of these foreign stations for comparison was due to the availability of sufficient information on the operation of their greenhouses in open-access literary sources.

**Table 4.** Micronutrient content of the leafy vegetable crops produced in a phytotechnical complex greenhouse at the Russian Antarctic station "Vostok" for five growing seasons.

| Indicators | Arugula *Eruca sativa* (Mill.) Thell., Cultivars | | Garden Cress *Lepidium sativum* L., Cultivars | | Chinese Cabbage *Brassica rapa* subsp. *Pekinensis* (Lour.) Hanelt cv. Vesnyanka | Japanese Cabbage *Brassica rapa* ssp. *nipposinica* (L.H.Bailey) Hanelt cv. Mizuna Green | Mustard *Brassica juncea* (L.) Czern. cv. Red Giant | Leaf Radish *Raphanus sativus* var. *Oleiformis* Pers. cv. Rax | Amaranth *Ama-ranthus* L. cv. Bagrya-nets | Leaf Lettuce *Lactuca sativa* L., Cultivars | | Parsley Leaf *Petro-selinum crispum* var. *crispum* Mill. cv. Bogatyr | *Dill Anethum graveolens* L. cv. Russian Giant |
|---|---|---|---|---|---|---|---|---|---|---|---|---|---|
| | Gurman | Barokko | Aghur | Vesenny | | | | | | Typhoon | Lollo Rossa | | |
| | | | | | | Raw ash | | | | | | | |
| % a.d.m.* | 19.3 ± 0.9 | 19.4 ± 1.3 | 24.0 ± 1.5 | 22.9 ± 2.4 | 22.3 ± 2.4 | 24.5 ± 0.9 | 23.3 ± 1.6 | 20.3 ± 3.2 | 27.2 ± 1.0 | 15.5 ± 1.7 | 19.7 ± 2.0 | 13.9 ± 0.5 | 19.4 ± 1.1 |
| coeff. var. | 5.3 | 7.7 | 7.3 | 11.9 | 12.0 | 10.1 | 4.3 | 18.0 | 4.3 | 12.3 | 11.6 | 4.2 | 6.4 |
| | | | | | | Mg content | | | | | | | |
| % a.d.m. | 0.57 ± 0.08 | 0.58 ± 0.11 | 0.66 ± 0.09 | 0.58 ± 0.03 | 0.60 ± 0.09 | 0.87 ± 0.15 | 0.61 ± 0.08 | 0.54 ± 0.10 | 0.90 ± 0.05 | 0.46 ± 0.06 | 0.47 ± 0.05 | 0.35 ± 0.03 | 0.38 ± 0.03 |
| coeff. var. | 15.5 | 21.7 | 15.7 | 5.3 | 16.6 | 19.7 | 15.6 | 21.1 | 6.3 | 15.1 | 12.5 | 8.9 | 10.2 |
| | | | | | | Fe content | | | | | | | |
| mg/kg a.d.m. | 57.22 ± 3.43 | 71.60 ± 7.19 | 63.92 ± 7.37 | 62.68 ± 2.31 | 59.92 ± 6.13 | 79.93 ± 20.13 | 72.7 ± 6.63 | 59.60 ± 13.16 | 91.44 ± 5.18 | 92.58 ± 8.58 | 98.24 ± 5.67 | 112.64 ± 6.80 | 98.68 ± 6.96 |
| coeff. var. | 6.8 | 11.5 | 19.3 | 4.2 | 11.7 | 28.7 | 10.4 | 25.2 | 6.5 | 10.6 | 6.6 | 6.9 | 8.1 |
| | | | | | | Mn content | | | | | | | |
| mg/kg a.d.m. | 48.00 ± 4.74 | 32.92 ± 6.31 | 48.98 ± 3.29 | 33.90 ± 2.46 | 64.18 ± 3.56 | 71.35 ± 9.42 | 27.64 ± 4.83 | 45.83 ± 2.86 | 46.20 ± 3.62 | 117.52 ± 5.58 | 167.18 ± 15.86 | 70.9 ± 5.46 | 141.88 ± 9.21 |
| coeff. var. | 11.3 | 21.9 | 7.7 | 8.3 | 6.3 | 15.1 | 19.9 | 7.1 | 8.9 | 5.4 | 10.8 | 8.8 | 7.4 |
| | | | | | | Cu content | | | | | | | |
| mg/kg a.d.m. | 6.58 ± 0.51 | 8.72 ± 0.72 | 9.27 ± 0.89 | 7.59 ± 0.28 | 5.41 ± 0.39 | 11.38 ± 0.51 | 8.12 ± 0.38 | 6.56 ± 1.22 | 8.46 ± 1.07 | 6.50 ± 0.56 | 5.21 ± 0.48 | 2.45 ± 0.35 | 8.13 ± 0.64 |
| coeff. var. | 8.81 | 9.5 | 11.0 | 4.2 | 8.1 | 5.1 | 5.3 | 21.2 | 14.4 | 9.8 | 10.6 | 16.4 | 8.9 |
| | | | | | | Zn content | | | | | | | |
| mg/kg a.d.m. | 43.36 ± 6.94 | 55.58 ± 4.55 | 116.4 ± 9.09 | 72.18 ± 4.26 | 85.38 ± 9.03 | 89.1 ± 4.30 | 77.1 ± 6.41 | 55.18 ± 2.65 | 74.34 ± 8.17 | 77.80 ± 4.82 | 103.54 ± 6.54 | 72.04 ± 7.52 | 130.74 ± 5.84 |
| coeff. var. | 18.3 | 9.3 | 8.9 | 6.7 | 12.1 | 5.5 | 9.5 | 5.5 | 12.5 | 7.1 | 7.2 | 11.9 | 5.1 |

Note: * a.d.m.—absolutely dry matter; the average values and confidence interval (M ± CI) at a 95% probability level were introduced in the table.

The results of the studies on the phytotechnical complex and greenhouses at these Antarctic stations are of particular interest due to the fact that the obtained experimental data can serve as a real basis for the development of future biological life support systems during crew flights to deep space, including to the Moon or Mars.

### 3.2. Quality and Safety of Plant Products

Evaluation of the key indicators of the quality and safety of plants grown in a phytotechnical complex greenhouse over five growing seasons showed that the nitrate content in plants (Figure 4A) was below the standard value, i.e., the maximum permissible concentration (MPC) for nitrates of the Russian Federation (2000 mg/kg wet weight of leafy vegetables crops) [24] and even lower than the MPC for nitrates accepted in other countries, with higher values than in Russia [25,26]. The content of dry matter (Figure 3C), microelements (Table 4), and vitamin C (Figure 4B) was similar to that in plants at the AFI agrobiopolygon.

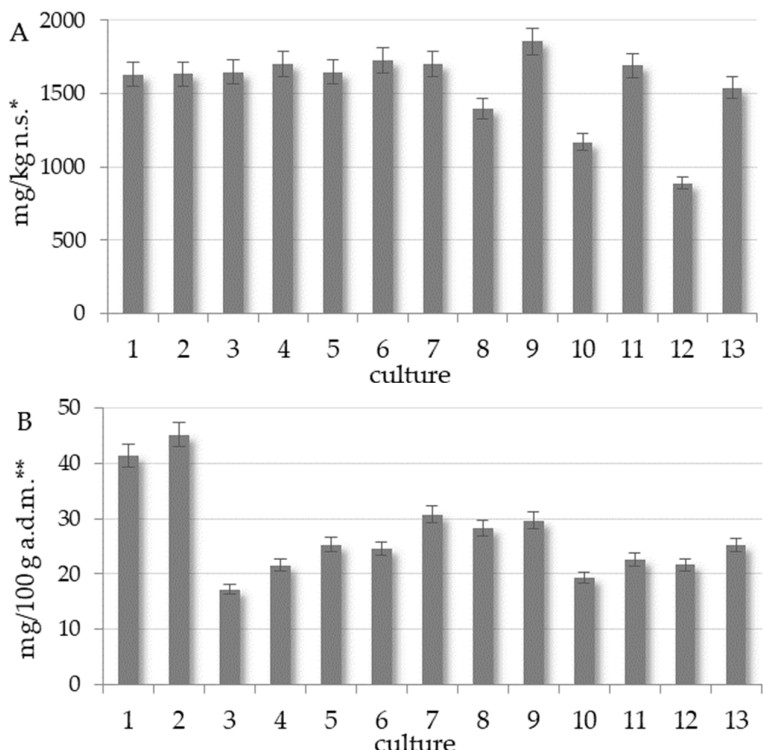

**Figure 4.** Content of nitrate (**A**) and vitamin C (**B**) of the studied vegetable crops grown in a phytotechnical complex at the Russian Antarctic station "Vostok" (five growing seasons); 1—Arugula *Eruca sativa* (Mill.) Thell., cv. Gurman; 2—Arugula *Eruca sativa* (Mill.) Thell., cv. Barokko; 3—Garden cress *Lepidium sativum* L. cv. Aghur; 4—Garden cress *Lepidium sativum* L. cv. Vesenny; 5—Chinese cabbage *Brassica rapa* subsp. *pekinensis* (Lour.) Hanelt, cv. Vesnyanka; 6—Japanese cabbage *Brassica rapa* ssp. Nipposinica (L.H.Bailey) Hanelt cv. *Mizuna Green*; 7—Mustard *Brassica juncea* (L.) Czern. cv. Red Giant; 8—Leaf radish *Raphanus sativus* var. *Oleiformis Pers.* cv. Rax; 9—*Amaranth Amaranthus* L. cv. Bagryanets; 10—Leaf lettuce *Lactuca sativa* L. cv. Typhoon; 11—Leaf lettuce Lactuca sativa L. cv. Lollo Rossa; 12—Parsley leaf *Petroselinum crispum* var. *crispum* Mill. cv. Bogatyr; 13—Dill *Anethum graveolens* L. cv. Russian giant; * n.s.—native substance, ** a.d.m.—absolutely dry matter. Bars show the average values and confidence interval (M ± CI) at a 95% probability level.

It should be noted that the content of dry matter measured for parsley and dill plants was the highest (Figure 3C), and the highest raw ash content (Table 4) was found in Japanese cabbage Mizuna and Amaranth Bagryanets cultivars. The values of the estimated indicators of the quality of plant production were quite even in repetitions; the coefficients of variation were mainly in the range of 5.0–22.0% and did not exceed 29% (Table 4). Taken

as a whole, the data indicate that the grown plant production was of high quality, that its safety was confirmed by the content of nitrates, which was less than the MPC of the Russian Federation and other countries [24–26], and the absence of pesticides and other pollutants in the growing system.

### 3.3. Psychological Aspect of the Plants' Presence at the Antarctic "Vostok" Station

The stable and high-productive functioning of the greenhouse at the "Vostok" station made it possible to grow plants not only for scientific research but also as a regularly produced food supplement with valuable nutritional properties that could be added to the standard diet. Fresh vegetable crops grown at the "Vostok" station provided a source of vitamins, minerals, essential amino acids, fiber, and other biologically active substances of natural origin, which are extremely necessary for polar researchers during their long stay at the station, increasing their immunity and maintaining their general biological tone and health.

Along with the nutritional function, the presence and cultivation of plants creates a favorable psychological and emotional environment. The model and biological experiments carried out by the IBMP on the "Mir" orbital space station and the "International Space Station" showed that under conditions of isolation, plants had a positive effect on the mental status of the crew, prevented the adverse effects of isolation factors (sensory deprivation, monotony), and contributed to the overall improvement of the psychological climate in the crew [7,27,28].

The "Vostok" Antarctic station, due to its geographical location and extreme conditions, is considered an analogue of a long-term habitable space base on the Moon, and the information and data obtained here make it possible to predict the behavior of people and their emotional state while inhabiting a space base. The staff of the "Vostok" station greatly appreciated the greenhouse and noted its importance in contributing to the sense of a normal life.

Thus, when evaluating the results of psychological surveys taken by the polar researchers at the beginning of the overwintering period, which asked about their expectations regarding the presence of plants at the station, it was revealed that none of the respondents expressed negative expectations. Out of 10 researchers, 1 noted a neutral stand and 9 expressed their expectations as "rather positive" or positive. Of the nine participants who expressed positive expectations, four noted the importance of the decorative, visual aspect of plants and three had expectations of the presence of a source of fresh food.

Analysis of the survey data taken at the end of the overwintering period, which evaluated the survey respondents' interaction with the plants during their stay at the station, showed that out of 12 respondents, 4 noted that they had discussed (talked about) plants with other participants "very often", 2 said "sometimes", and 6 said almost never or never. Five out of twelve participants noted that they preferred to spend time near the plants "often or very often", two selected "sometimes", and five subjects spent almost no time near the plants.

Changes in the state of the plants were also noted by 7 out of 12 subjects, with 5 of them selecting the answer "often or very often" and 2 choosing "sometimes". These 7 respondents noted the psychological effects of plants as positive (from "rather positive" to "very positive"), 4 of the remaining participants noted no effect on mood, and 1 noted a negative effect, despite the declared absence of interactions with the plants.

Thus, the absolute majority (90%) of respondents at "Vostok" station expressed positive expectations from the presence of plants at the station, and 40% of respondents noted the importance of the decorative aspect of plants.

At the end of overwintering, about 60% of the polar researchers pointed to the positive psychological effect of plants and noted a preference to spend time near them. The nature of the human–plant interaction at the station was partly related to the researchers' previous daily experienced interactions with house plants, as well as the nature of their expectations about the presence of plants in the human accommodation of the Antarctic station.

## 4. Discussion

The subject of this study—growing plants, for example, leaf vegetable crops, in the Antarctic in close proximity to humans—remains relevant due to the high physiological and psychological needs of polar researchers living under extreme environmental conditions and isolated from social life and the natural ecobiome. Additionally, various countries' scientific communities and organizations show much interest in the creation, testing, and implementation of original systems for year-round, waste-free plant production at Antarctic stations.

Comparative studies conducted by AFI, AARI, and IBPM in analogous phytotechnical complex greenhouses at "Vostok" station and the AFI agrobiopolygon confirmed the hypothesis of this study and allowed for the identification of differences in the response of the same cultivated crops to environmental conditions differing in barometric pressure and partial pressure of oxygen. Obviously, these differences are due to the well-known fact that the Antarctic "Vostok" station is located at 3488 m above sea level (high mountain conditions), and the AFI agrobiopolygon in St. Petersburg is at 28 m above sea level.

The tested lettuce crops turned out to be resistant to the reduced values of barometric pressure and partial pressure of oxygen at "Vostok" station, while cabbage leaf crops, parsley, dill, and amaranth showed lower green mass yields compared to those at the AFI agrobiopolygon. It can be assumed that such a difference in the reaction of lettuce and other crops to environmental conditions is to some extent related to their origin and the resulting differences in the adaptability of plants to conditions in high mountains. At the same time, it is interesting to note the similarity in the quality and safety indicators in plants at the "Vostok" station and at the AFI agrobiopolygon.

The similar value in edible part yield per day but different value in electricity and water consumption noted in the comparative evaluation of the effectiveness of Antarctic greenhouses may be related to the following factors: Higher electricity consumption in the greenhouse at the German station is due to the fact that the greenhouse is located in a separate structure and its operation is more costly, taking into account the climate in Antarctica [22]. At the U.S. station, it is mainly due to the use of more energy-consuming, water-cooled 1000 W high-pressure lamps as light sources [6,22]. It should be noted that the productivity of the Russian phytotechnical complex, exactly the same as that of the greenhouses at the mentioned foreign Antarctic stations [22], was at least 1.5–2 times higher than that of modern greenhouse complexes in the European Union and other countries with controlled conditions and supplementary artificial lighting [28,29].

The results of the assessment of the emotional and psychological attitudes towards plants among polar researchers clearly showed that the nature of their interaction with plants at the station was partly related to the characteristics of their previous daily experience interacting with house plants, as well as the nature of their expectations regarding the presence of plants in the wintering grounds. With the passage of time at the station, most polar researchers felt increasing sympathy and desire to see plants growing in the phytotechnical complex. The same was noted among cosmonauts and other testers who were in long-term isolation [9,27,30]. The psychological benefits of indoor plants, especially in windowless and confined environments, were also described previously in a number of studies [31,32].

## 5. Conclusions

Comparative studies of the growth and development of some leafy vegetable crops, looking at the yield and quality of their production when grown in an original phytotechnical complex greenhouse at the Antarctic station "Vostok" and at the AFI agrobiopolygon (Russia, St. Petersburg), have revealed differences in the "genotype–environment" interaction in plants. Lettuce crops turned out to be the most adapted to the growing conditions at the "Vostok" Antarctic station, while other crops (cabbage leaf crops, amaranth, dill, leaf parsley) had significantly lower yields of their mass compared to those at the AFI agrobiopolygon, which may be due to the difference in the values of barometric pressure and partial pressure of oxygen at these areas.

Nevertheless, in terms of yield and quality per unit area and time, the studied plants in the phytotechnical complex greenhouse located in the living room of the station "Vostok" showed similar values to those in greenhouses located in separate rooms or structures at some Antarctic stations, according to published data. At the same time, the consumption of electricity and water per unit of production were lower at the phytotechnical complex greenhouse at the Antarctic station "Vostok". In terms of quality and safety, the plant production obtained fully met the basic sanitary and hygienic requirements of the Russian Federation and other countries.

Along with their nutritional function, plants in the phytotechnical complex greenhouse had a significant positive psychological and emotional effect on polar researchers.

The data obtained from testing the phytotechnical complex at the Antarctic station "Vostok" indicate the need to continue research aimed at selecting varieties and hybrids of leaf vegetable crops which are most adapted to the growing conditions at this station.

**Author Contributions:** Conceptualization, Y.V.C. and E.A.I.; Methodology, G.G.P., T.E.K., A.V.T., D.M.S. and Y.V.K.; Software, G.V.M.; Validation, G.G.P., M.A.L. and E.A.I.; Formal Analysis, G.V.M. and E.V.K.; Investigation, A.V.T., A.B.N., O.R.U., G.G.P., T.E.K. and Y.V.K.; Resources, Y.V.C. and M.A.L.; Data Curation, G.G.P., G.V.M. and M.A.L.; Writing—Original Draft Preparation, G.G.P., E.V.K., D.M.S., T.E.K. and M.A.L.; Writing—Review and Editing, Y.V.C., E.A.I., M.A.L. and G.G.P.; Visualization, A.V.T., A.B.N. and O.R.U.; Supervision, Y.V.C. and E.A.I.; Project Administration, Y.V.C. All authors have read and agreed to the published version of the manuscript.

**Funding:** This research did not receive external funding.

**Data Availability Statement:** The data presented in this study are available upon reasonable request from the corresponding author.

**Acknowledgments:** We express our deep gratitude to the direction of the AARI, IBMP, and AFI for the opportunity to implement the project for the creation of a phytotechnical greenhouse complex and its testing at the "Vostok" Russian Antarctic station simultaneously with that at the AFI agrobiopolygon, as well as to all the engineers and industrial partners who took part in the development and/or manufacture of the phytotechnical complex greenhouse.

**Conflicts of Interest:** The authors declare that the research was conducted in the absence of any commercial or financial relationships that could be construed as a potential conflict of interest.

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
