# Peer review of "Growth and Development of Leaf Vegetable Crops under Conditions of the Phytotechnical Complex in Antarctica"

_agronomy, doi:10.3390/agronomy13123038_

Round 1

Reviewer 1 Report

Comments and Suggestions for Authors

Evaluation of the manuscript agronomy-2719740-peer-review-v1:

GROWTH AND DEVELOPMENT OF LEAF VEGETABLE CROPS UNDER CONDITIONS OF THE PHYTOTECHNICAL COMPLEX IN ANTARCTICA

Authors: Gayane G. Panova1*, Andrey V. Teplyakov2, Anatoliy B. Novak2, Margarita А. Levinskikh3, Olga R. Udalova1, Yuriy  V. Khomyakov1, Evgeniy A. Ilyin3, Yuriy V. Chesnokov1

1 Agrophysical Research Institute (AFI), 195220 Saint-Petersburg, Russia

2 Arctic and Antarctic Research Institute (ААRI), 199397 Saint-Petersburg, Russia

3 Institute of Biomedical problems (IBMP), 123007 Moscow, Russia

* Correspondence: gaiane@inbox.ru; Tel.: +7(812)535-79-09

General assessment of the research carried out and the results obtained:

The scientific study, carried out on the 13 varieties of 9 leafy vegetable crops, cultivated in two locations with different climatic conditions, respectively at the Russian Antarctic station "Vostok" and Agrophysical Research Institute (AFI), emphasized at a high scientific level the genotype-environment interaction , particularly important aspect of growing a plant under economic efficiency conditions

(high productions per unit area with low costs).

The results regarding the quality of green mass and dry matter production, showed values within the standard limits, being very well documented from a scientific point of view with results obtained in the specialized literature.

The importance of research on quality chemical analysis, carried out on leafy vegetable crops grown at the "Vostok" station, is essential in evaluating the source of vitamins, minerals, essential amino acids, fibers and other biologically active substances of natural origin, for a healthy diet of polar researchers during their long stay at the station.

The importance of the research initiated in this experiment is supported at a high scientific level by the 34 bibliographic titles that are mentioned in the text, of which 16 titles in the last 5 years and 24 in the last 10 years, which reveal scientific news in the field, which are related with the objective of the research addressed in this paper.

Scientific novelty:

The results of research carried out at the Antarctic station "Vostok" present a starting point for the coexistence of humans with plants in a controlled environment, contributing to the development of future biological life support systems during flights to the Moon or Mars.

Observations on materials and methods:

The technical presentation is not clear enough:

The number of plants resulting from the quantity of 1 g of seed of leafy vegetable crops, sown on 0.02 m2 (20 cmx10 cm) is not specified.... Example: theoretically 1 gram of seed: lettuce (Lactuca sativa) contains over 1000 seeds (reference: How to grow lettuces - Sea Spring Seeds); in radishes, 1 gram contains approx. 100 seeds (reference:  SciELO - Brazil - Seed quality and optimal spatial arrangement of fodder radish Seed quality and optimal spatial arrangement of fodder radish  )….. which is exaggerated for the surface of 0.02 m2!!!

I believe that it is necessary to mention (for each species of vegetable plant) the number of plants harvested on the surface of 0.02 m2, in order to evaluate the production results over the 5 seasons (table no. 3).

I suggest to mention how the coefficients of variation are calculated for the indicators of growth and development of the studied vegetable crops?..(according to the productions of the 5 seasons?…. or… according to the number of repetitions?) ...!! !!

I also suggest that the number of repetitions be clearly stated (two repetitions?..or three repetitions?), since the text generally mentions 2-3 repetitions.

REVIEWER 1

Author Response

Response to Reviewer 1 Comments

Dear Reviewer! We are thankful for reviewing our manuscript and for the constructive comments.

Comment 1

  1. The number of plants resulting from the quantity of 1 g of seed of leafy vegetable crops, sown on 0.02 m2 (20 cmx10 cm) is not specified.... Example: theoretically 1 gram of seed: lettuce (Lactuca sativa) contains over 1000 seeds (reference: How to grow lettuces - Sea Spring Seeds); in radishes, 1 gram contains approx. 100 seeds (reference: SciELO - Brazil - Seed quality and optimal spatial arrangement of fodder radish Seed quality and optimal spatial arrangement of fodder radish )….. which is exaggerated for the surface of 0.02 m2!!!

Response: The increased seed sowing density is justified by the technology of growing plants on thin-layer panoponics. Each 0.02 m2 = 50 cm x4 cm. The number of seeds is calculated in grams.

An explanation was given in lines: 196-198: «Sowing seeds of each crop was carried at the density of 1 g/0.02 m2 (repetition), which was repeated in each growing season twice for most cultivated crops, for arugula - three times. Each crop was grown over five growing seasons during 2020-2021».

Comment 2

  1. I believe that it is necessary to mention (for each species of vegetable plant) the number of plants harvested on the surface of 0.02 m2, in order to evaluate the production results over the 5 seasons (table no. 3).

Response: The number of plants for each type of vegetable crop was not taken into account, since the sowing was dense. Some analogies can be seen when growing microgreens.

Comment 3

  1. I suggest to mention how the coefficients of variation are calculated for the indicators of growth and development of the studied vegetable crops?..(according to the productions of the 5 seasons?…. or… according to the number of repetitions?) ...!! !!

Response: Over the course of two years, 5 growing seasons were carried out for each crop. In our studies, the repetition was sowing seeds weighing 1 g on an area of 0.02 m2. In each growing season, arugula was sown in 3 replicates, other crops - in 2 replicates. The coefficients of variation of growth and development indicators of the studied vegetable crops were calculated by the number of repetitions over five growing seasons.

Comment 4

  1. I also suggest that the number of repetitions be clearly stated (two repetitions?..or three repetitions?), since the text generally mentions 2-3 repetitions.

Response: we made an explanation in lines: 196-198: «Sowing seeds of each crop was carried at the density of 1 g/0.02 m2 (repetition), which was repeated in each growing season twice for most cultivated crops, for arugula - three times. Each crop was grown over five growing seasons during 2020-2021».

Reviewer 2 Report

Comments and Suggestions for Authors

I read with interest the manuscript entitled “Growth and Development of Leaf Vegetable Crops under Conditions of the Phytotechnical Complex in Antarctica”. The purpose of this work was to study the growth and development of leaf vegetable crops and the main quality indicators of their edible parts when grown in the phytotechnical complex-greenhouses at the “Vostok” Antarctic station and at the agrobiopolygon of the Agrophysical Research Institute. The subject of the article is important and has great relevance for the scientific environment of the study area. Therefore, the manuscript needs some adjustments so that it can then be forwarded to the publication process. The manuscript has the potential for publication in this journal Agronomy and needs the following adjustments:

ABSTRACT

- Insert an introductory section on the subject. The authors started by talking about the objectives.

- The objective must be the same as mentioned in the Introduction.

- Mention which cultures were evaluated in the research.

- There are many keywords, review the newspaper's rules to see if it is possible.

- Replace the keywords that are present in the title.

INTRODUCTION

- There is a lot of information cited that needs reference, for example the first paragraph. Review this throughout the text.

- Hypotheses must be affirmative and not suppositions. To review.

MATERIAL AND METHODS

- Is Figure 1 specific only to the study region? These light absorption spectra are similar to what is found in the literature for sunlight for all plants.

- Why was this questionnaire carried out with people? Did these results enter the work? Was any consent form signed by these people? See if this analysis is really important for this work.

RESULTS AND DISCUSSION

- If possible, replace Tables with Figures. Doing it this way makes it better for readers to understand the results.

- Use some statistical test to compare means.

- Only means and confidence intervals are not recommended when wanting to compare means.

- With this new analysis, most of the results will be modified.

CONCLUSION

  - Very extensive conclusion. Reduce.

Author Response

Response to Reviewer 2 Comments

Dear Reviewer! We are thankful for reviewing our manuscript and for the constructive comments.

ABSTRACT

Comment 1

  1. Insert an introductory section on the subject. The authors started by talking about the objectives.

 Response: it was done.

Comment 2

  1. The objective must be the same as mentioned in the Introduction.

 Response: It was done

Comment 3

  1. Mention which cultures were evaluated in the research.

 Response: it was done.

Comment 4

  1. There are many keywords, review the newspaper's rules to see if it is possible.

 Response: it was done.

Comment 5

  1. Replace the keywords that are present in the title.

Response: it was done.

INTRODUCTION

Comment 6

  1. There is a lot of information cited that needs reference, for example the first paragraph. Review this throughout the text.

Response: it was done.

Comment 7

  1. Hypotheses must be affirmative and not suppositions. To review.

 Response: it was done.

MATERIAL AND METHODS

 Comment 8

  1. Is Figure 1 specific only to the study region? These light absorption spectra are similar to what is found in the literature for sunlight for all plants.

Response:

The figure 1 shows the experimentally recorded spectrum of the LED lamp. The lamp simulates sunlight in the visible range with a color temperature of 4000 K, which corresponds to morning light when the sun is at an angle of about 20 degrees relative to the horizon.

The sentence “The spectrum of light in the PAR region is close to that of sunlight. White LEDs are employed with modified secondary optics using a polymer luminophor (know-how).” at the lines: 174, 175, - was replaced with sentence at the lines: 172-176 “Light sources modulating sunlights spectrum in the photosynthetically active radiation (PAR) were used. The white LEDs included in the lamps with modified secondary optics using a polymer phosphor (know-how) with a color temperature of 4000K simulated morning sunlight, when the sun is at an angle of about 20 degrees relative to the horizon”.

Comment 9

  1. Why was this questionnaire carried out with people? Did these results enter the work? Was any consent form signed by these people? See if this analysis is really important for this work.

Response:

The questionnaire/survey was carried out in order to determine psychological value of the greenhouse for Vostok station personnel. According to available literary sources, it is well established that presence of plants in living and working spaces, especially in window less, isolated and confined conditions, can significantly improve mood and subjective quality of life. Thus, we consider positive psychological effects of the greenhouses among main advantages of their use in Antarctic conditions, along with nutritional and other qualities of the plants.

The subjects signed informed consent forms for participation in all medical and psychological studies at Vostok station. The studies were approved by Bioethics Committee of the Institute of Biomedical Problems, Russian Academy of Sciences.

Some corrections and additions were added to the manuscript: e.g., lines: 219-225, 514-516.

2. literary sources were added (31, 32).

RESULTS AND DISCUSSION

Comment 10

  1. If possible, replace Tables with Figures. Doing it this way makes it better for readers to understand the results.

 Response: it was done.

Comment 11

  1. Use some statistical test to compare means.

 Response: it was done.

Comment 12

  1. Only means and confidence intervals are not recommended when wanting to compare means.

 Response: it was done. A Duncan’s multiple range test was used to determine the significance of differences between mean values.

Comment 13

  1. With this new analysis, most of the results will be modified.

 Response: Results adjusted.

CONCLUSION

Comment 14

  1. Very extensive conclusion. Reduce.

 Response: it was done.

Reviewer 3 Report

Comments and Suggestions for Authors

Dear Authors,

I would like to make the following comments regarding the manuscript:

The choice of title and abstract are adequate, but the number of keywords is too many. I ask the authors to use only the most relevant ones.

The literature references used are adequate and a large number of them have been published in the last 5 years and are related to the MDPI principles. The Introduction chapter is more than two pages long. This is what I ask the authors to shorten and compress the information.

In the Results chapter, the intertextual references of the tables and figures are not followed by the figures and tables themselves. Please the authors fix this.

There are typos and mistakes in many places, so it's worth re-reading the text with this eye on it.

The format is correct, the Discussion and Conclusions are correct.

Comments on the Quality of English Language

English is prone to minor errors, some of which are typos and inaccuracies. These need to be corrected. 

Author Response

Response to Reviewer 3 Comments

Dear Reviewer! We are thankful for reviewing our manuscript and for the constructive comments.

Comment 1

  1. The choice of title and abstract are adequate, but the number of keywords is too many. I ask the authors to use only the most relevant ones.

 Response: it was done.

Comment 2

  1. The Introduction chapter is more than two pages long. This is what I ask the authors to shorten and compress the information.

Response:

In the introduction, we tried to present in a concise form information about the existing main types of greenhouses at Antarctic stations, the features of their placement and, if it was known, about the techniques and methods of growing plants on them. This information was need to ultimately be more clear what their difference from the phytotechkompexes device at the Antarctic Vostok station. The material is presented in a very condensed form and any reduction will lead to some loss of comparative information.

Comment 3

  1. In the Results chapter, the intertextual references of the tables and figures are not followed by the figures and tables themselves. Please the authors fix this.

Response: it was done.

Comment 4

  1. There are typos and mistakes in many places, so it's worth re-reading the text with this eye on it.

Response: it was done.

Round 2

Reviewer 2 Report

Comments and Suggestions for Authors

The authors revised the manuscript, however, several things needed to be corrected.

Insert the title of the X axis of all graphs.

It was mentioned that the mean comparison test was added. The test has not been added. I suggest the addition once again.

Author Response

Response to Reviewer 2 Comments

Dear Reviewer! We are thankful for comments.

Comment 1

Insert the title of the X axis of all graphs.

Response: it was done.

Comment 2

It was mentioned that the mean comparison test was added. The test has not been added. I suggest the addition once again.

 Response: it was done. A Duncan’s multiple range test was used to determine the significance of differences between mean values.

Data of productivity (green mass yield per year) were analyzed by applying one-way analysis of variance (ANOVA) and a Duncan’s multiple range test was used to determine the significance of differences between mean values.

 An explanatory text about our use of three methods for analyzing averages has been added to the subsection Statistical analysis of the section 2. Materials and Methods.

Lines 230-237: The mean values of the studied parameters and their confidence intervals (CI) M ± CI were determined. One-way analysis of variance (ANOVA) yield of leafy vegetable crops grown in phytotechnical complex at arctic station "Vostok" and agrobiopolygon of AFI and Duncan’s multiple range test was used to determine the significance of differences between mean values. Differences between options were considered significant at p ≤ 0.05. When assessing the content of micronutrients in leafy vegetable crops grown during 5 growing seasons at the Russian Antarctic “Vostok” Station yield variability was assessed using the coefficient of variation CV [21].

 A clarification has been added in the note below Table 2.

Lines 302-305: Note: The table shows the average values and confidence interval of plant yield (M ± CI) at a 95% probability level. Data are presented as the means of five experiments with three biological replications per variant for arugula Eruca  sativa (Mill.) Thell., cvs. Gurman, Barokko and means of five experiments with two biological replications per variant for other studied crops. Values in columns followed by different letters (a–b) are significantly different at p ≤ 0.05, as determined by Duncan’s multiple range test.

 The Diagram in Figures 3 and 4 shows average values and confidence interval of crop yield (M ± CI) at a 95% probability level.

Lines 344, 345 and 423-425: It was added “…Bars show the average values and confidence interval (M ± CI) at a 95% probability level..”

 Lines 431: The average values and confidence interval (M ± CI) at a 95% probability level were introduced in the table (4).
